# Multimodal Pathway: Improve Transformers with Irrelevant Data from Other Modalities

## Abstract

We propose to improve transformers of a specific modality with irrelevant data from other modalities, e.g., improve an ImageNet model with audio or point cloud datasets. We would like to highlight that the data samples of the target modality are irrelevant to the other modalities, which distinguishes our method from other works utilizing paired (e.g., CLIP) or interleaved data of different modalities. We propose a methodology named Multimodal Pathway: given a target modality and a transformer designed for it, we use an auxiliary transformer trained with data of another modality and construct pathways to connect components of the two models so that data of the target modality can be processed by both models. In this way, we utilize the universal sequence-to-sequence modeling abilities of transformers obtained from two modalities. As a concrete implementation, we use a modality-specific tokenizer and task-specific head as usual but utilize the transformer blocks of the auxiliary model via a proposed method named Cross-Modal Re-parameterization, which exploits the auxiliary weights without any inference costs. We observe significant and consistent performance improvements with irrelevant data of image, point cloud, video, and audio. For example, on ImageNet-1K, a point-cloud-trained auxiliary transformer can improve an MAE-pretrained ViT by 0.6% and a ViT trained from scratch by 5.4%. The code and models will be publicly available.

## 1 Introduction

Transformers are widely adopted in various tasks across modalities, such as text classification (Devlin et al., 2019), object detection (Carion et al., 2020), point cloud analysis (Zhao et al., 2021), and audio spectrogram recognition (Gong et al., 2021a). Apart from numerous unimodal tasks, transformers are also effective on multimodal data, *e.g.*, CLIP (Radford et al., 2021) uses image-text pairs to achieve superior performance in image recognition. Transformers' success in multiple modalities demonstrates their abilities to universally establish sequence-to-sequence modeling, given the input sequences (*i.e.*, tokens) which can be seen as the universal embeddings of data of multiple modalities (Dosovitskiy et al., 2021; Carion et al., 2020; Xie et al., 2021; Zhao et al., 2021; Gong et al., 2021a; Wang et al., 2022b). For brevity, we refer to such ability as the *universal modeling ability*.

We would like to note that CLIP (Radford et al., 2021) represents the significant success of a methodology that improves a model's performance on a certain modality (*i.e.*, image) with the help of data from another modality (*i.e.*, text), but the limitation is also apparent - **the data samples from the two modalities must be relevant** (*i.e.*, paired). This limitation seems so inevitable that it hardly attracts research interest from the literature. Taking another two modalities, image and audio, as an example, we may expect that training with image-audio pairs may help the model recognize images (if we build a dataset with enough image-audio pairs and re-design the model to use the audio labels as the supervision, just like CLIP does with image-text pairs), but **it seems hard to believe that a pure audio dataset would improve a model's performance on ImageNet classification without any relevance between the audio and image samples**.

In this paper, we propose to improve a transformer's performance on a certain modality even with irrelevant data from another modality. The motivation is that we can see a training process on a certain modality as converting the data of the modality to sequences (*i.e.*, tokens) and establishing

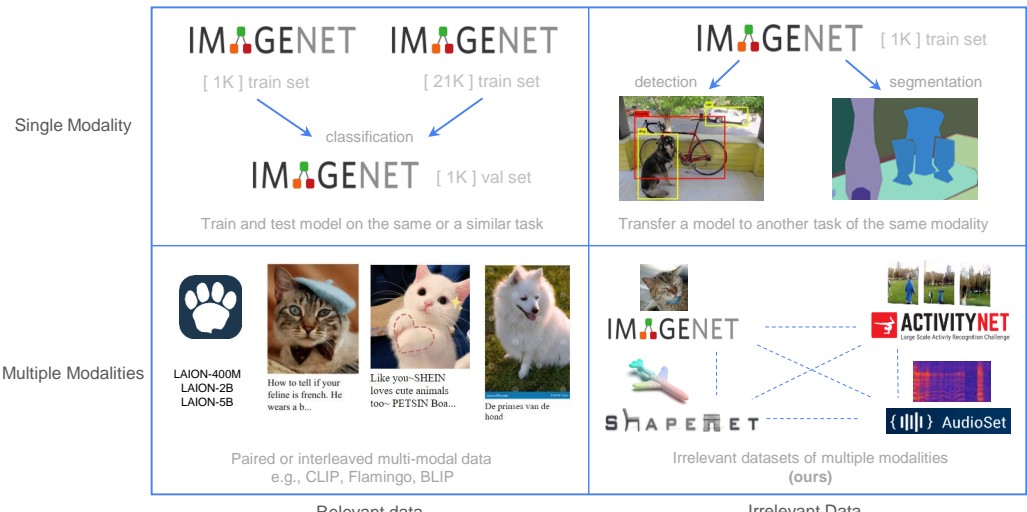

Figure 1: Compared with the known paradigms, we focus on scenarios where the data samples are from multiple modalities but irrelevant, which is an open problem in the literature.

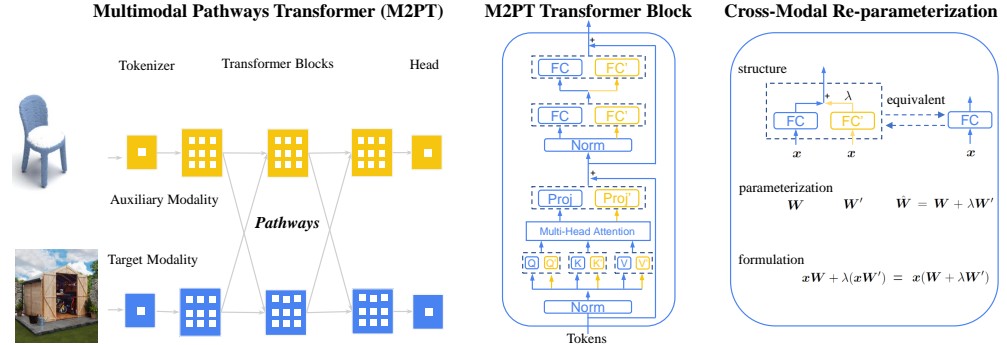

Figure 2: (**Left**) The framework of Multimodal Pathway Transformer (M2PT). We take point cloud and image as the two representative modalities. Common practices with transformers follow the same pipeline: using 1) tokenizers to convert the input data to sequences, 2) transformer blocks to process the sequences, and 3) heads to decode the sequences. We upgrade the sequence-to-sequence modeling by establishing *pathways* between the components of different modalities so that processing the tokens of a specific modality can utilize the transformer blocks trained with another modality. (**Middle**) Our design of M2PT, where the pathways are implemented by letting a linear layer (including the Query/Key/Value/projection layers in the attention block and those in the FFN block) in the target model cooperate with its counterpart in the auxiliary model. (**Right**) Cross-Modal Re-parameterization efficiently realizes M2PT by re-parameterizing the weights of the target model with those of the auxiliary model, which introduces completely no inference costs.

sequence-to-sequence mappings with the transformer blocks. For a specific modality, we reckon that the trained model has knowledge encoded in the sequence-to-sequence modeling that can facilitate another modeling process whose input sequences are obtained from another modality. In other words, apart from the obvious modality-specific knowledge acquired through training on a specific modality, we seek the **modality-complementary knowledge of sequence-to-sequence modeling in transformers** and will show that **it does exist**.

However, given a target modality, it seems difficult to design the model to utilize some irrelevant data of another modality because the data samples of different modalities (e.g., image and audio) may vary significantly in the semantics, data format, preprocessing, and it seems hardly possible to design a reasonable objective function since there is no relevance between any two samples. In this paper, we solve this problem by not directly mixing training data of two modalities but *seeing a model trained on a specific unimodal dataset as a proxy of the corresponding modality*

*and using the model instead.* Specifically, given a target modality and an auxiliary modality, we propose a framework named *Multimodal Pathway* to improve the performance on the target modality by *using two transformers respectively trained with the unimodal data of the two modalities*. We construct *pathways* across the components of the target and auxiliary models to exploit the modality-complementary knowledge encoded in the latter to help the former. Note that pathway is an abstract concept that may refer to any connection between the two models. We name such a model as **M**ulti**m**odal **P**athway **T**ransformer (**M2PT**) for brevity.

This paper proposes a simple yet effective implementation of M2PT, where the key is the concrete implementation of pathways that connect the two models. As discussed above, thanks to the universal modeling ability, transformers on different modalities may have different tokenizers, but their main bodies (*i.e.*, transformer blocks) may have the same structure. For a target model and an auxiliary model with the same structure of the main bodies, a layer in the main body of the former should have a counterpart in the latter. For example, the counterpart of the Query layer in the 9th block of the target model, which is the 9th Query layer in the auxiliary model, should exist, and they play a similar role in the two models. Considering this, we build the connections between the two models by augmenting every linear layer in the transformer blocks of the target model with its counterpart in the auxiliary model. We let the two layers take the same inputs and add up their outputs, as shown in Fig. 2 (middle).

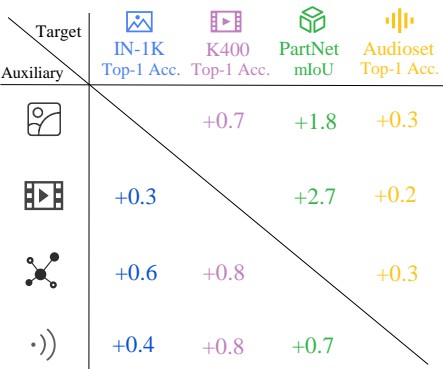

Figure 3: Multimodal Pathway Transformer brings consistent improvements on 4 modalities - image, video, point cloud, and audio.

However, considering the budget on compute and latency, we desire an implementation of the Multimodal Pathway that realizes the pathways and makes good use of the auxiliary model but does not increase the inference cost, compared to a regular model trained on the target modality. We would like to note that the structure described above can be equivalently implemented by a re-parameterization method, which equivalently converts the connections between model structures (*i.e.,* linear layers) into connections between the two models' weights. Specifically, we construct a pathway for each target linear layer by adding the corresponding weights of its counterpart in the trained auxiliary model scaled by a learnable multiplier that indicates the strength of the pathway, so that the method is named *Cross-Modal Re-parameterization*. A significant strength of re-parameterization is that we can merge the weights after training so that the structure and number of parameters of the resultant model will be identical to a regular model.

We conduct experiments across the image, video, point cloud, and audio modalities. Figure 3 shows that M2PT brings consistent improvements among 4 modalities. In specific, with a base-scale transformer, M2PT achieves 83.9% (+0.6) top-1 accuracy on ImageNet-1K, 82.3% (+0.8) on Kinectis-400, 47.6% (+2.7) mIoU on PartNet, and 35.6% (+0.3) on Audioset. Such results demonstrate that M2PT effectively improves transformers with irrelevant data from other modalities.

In summary, our contributions are as follows:

- We propose Multimodal Pathway, which is a framework to improve transformers via exploiting models trained on other modalities.
- We propose an inference-cost-free implementation of Multimodal Pathway, which is named Cross-Modal Re-parameterization.
- Multimodal Pathway represents an early exploration in this direction, which offers a novel perspective. We realize significant and consistent improvements on four representative modalities, which demonstrates the potential of our method as a promising approach.

## 2 METHOD

## 2.1 ARCHITECTURAL DESIGN

We design a transformer for a specific modality as three modules - the modality-specific tokenizer, the modality-agnostic transformer blocks, and the modality-specific head. We assume the dimension of tokens is $D$, which is a pre-defined architectural hyper-parameter, and describe how to tokenize the input data of multiple modalities.

**Image tokenizer**. Let's consider an image represented by $\boldsymbol{x} \in \mathbb{R}^{H \times W \times C}$, where $(H, W)$ specifies the image's resolution, and $C$ is the channel count. We transform this image into a series of 2D patches, denoted by $\boldsymbol{x}_p$, with dimensions $\mathbb{R}^{N_s \times (S^2 \cdot C)}$. In this representation, $S$ is the size of each patch, and $N_s$ (calculated as $HW/S^2$) indicates the total number of these patches. Following this transformation, we employ a projection layer to adjust the embedding dimension to $D$.

$$\boldsymbol{x}_I \in \mathbb{R}^{C \times H \times W} \to \boldsymbol{x}_I' \in \mathbb{R}^{N_s \times (S^2 \cdot C)} \to \boldsymbol{x}_I'' \in \mathbb{R}^{N_s \times D} \,. \tag{1}$$

**Video tokenizer**. Analogous to 2D images, we utilize video patches as the basic unit for learning video representations. Given a video $\boldsymbol{x} \in \mathbb{R}^{T \times C \times H \times W}$, we use the same patch size $S$ so that the video can be reshaped to $\boldsymbol{x} \in \mathbb{R}^{N_s' \times (S^2 \times C)}$, where $N_s' = \frac{T \times H \times W}{S^2}$. Upon deriving the video token sequence, a projection layer is also applied to obtain the video embeddings.

$$\boldsymbol{x}_V \in \mathbb{R}^{T \times C \times H \times W} \to \boldsymbol{x}_V' \in \mathbb{R}^{N_s' \times (S^2 \cdot C)} \to \boldsymbol{x}_V'' \in \mathbb{R}^{N_s' \times D} \,. \tag{2}$$

**Point cloud tokenizer**. Consider a point cloud $\mathcal{X} = \{\boldsymbol{x}_i\}_{i=1}^P$ comprising $P$ points. Each point $\boldsymbol{x}_i$ is defined as $\boldsymbol{x}_i = (\boldsymbol{p}_i, \boldsymbol{f}_i)$, where $\boldsymbol{p}_i \in \mathbb{R}^3$ denotes the 3D coordinates and $\boldsymbol{f}_i \in \mathbb{R}^c$ represents the feature of the $i$-th point. Typically, $\boldsymbol{f}_i$ encompasses visual cues such as color, viewpoint, normal, and so on. We utilize the Farthest Point Sampling (FPS) technique to sample a representative skeleton from the original point clouds at a fixed sampling ratio of 1/4. Subsequently, the $K$-Nearest Neighbor (KNN) method is employed to group proximate points. Leveraging grouped sets that encapsulate local geometric priors, we craft an adjacency matrix centered on the grouped subsets, aiming to extract the intricate structural details of 3D objects and scenes. Ultimately, the structural representations from $K$ subsets are aggregated. In essence,

$$\boldsymbol{x}_P \in \mathbb{R}^{P \times (3+c)} \to \boldsymbol{x}_P' \in \mathbb{R}^{\frac{P}{4} \times \frac{D}{2}} \to \boldsymbol{x}_P'' \in \mathbb{R}^{\frac{P}{16} \times D} \,. \tag{3}$$

**Audio spectrogram tokenizer**. Initially, we preprocess an audio waveform of duration $t$ seconds using the log Mel filterbank (Schneider et al., 2019). We then apply the Hamming window with a stride of $t_s$ on the frequency $f_s$, segmenting the original wave into $l = \frac{t}{t_s}$ intervals, thereby transforming the wave into an $l$-dimension filterbank. The spectrogram is subsequently divided into patches across both time and frequency dimensions, each of size $S$. Notably, unlike image patches, audio patches exhibit overlap on spectrograms. We partition the entire spectrogram into $N_s = 12 \left\lceil \frac{100t-16}{10} \right\rceil$ patches using a $S \times S$ (Denote in Eq. 1) convolution, and then flatten these patches into token sequences. Given $T$ and $F$ as the time and frequency dimensions respectively, the procedure can be summarized as

$$\boldsymbol{x}_A \in \mathbb{R}^{T \times F} \to \boldsymbol{x}_A' \in \mathbb{R}^{N_s \times S \times S} \to \boldsymbol{x}_A'' \in \mathbb{R}^{(N_s \cdot D/S^2) \times D} \,. \tag{4}$$

**Transformer blocks**. We simply adopt the structural design of the transformer blocks in Vision Transformer (ViT) (Dosovitskiy et al., 2021), where each transformer block comprises a self-attention block and a Feed-Forward Network (FFN) block. The linear layers include the Query/Key/Value/projection layers in the attention block and two layers in the FFN block. For fairness and reproducibility, we use the same architectural hyper-parameters (e.g., dimension of tokens, number of blocks, and number of heads) as ViT-Base for every M2PT model on every modality.

## 2.2 CROSS-MODAL RE-PARAMETERIZATION

For an M2PT model on a specific modality, we use Cross-Modality Re-parameterization in the transformer blocks to utilize another model's weights trained on another modality. Specifically, let $\theta$ be an arbitrary trainable parameter of a layer in the transformer, $x$ be the input, $y$ be the output, we use $f$ to denote the operation so that $y = f(x; \theta)$. With Cross-Modality Re-parameterization, we simply re-parameterize the layer with parameters of its counterpart in another modal that are trained on another modality. Let $\theta'$ be the parameter of the counterpart, the operation becomes

$$y = f(x; \theta + \lambda \theta') \,. \tag{5}$$

We refer to $\lambda$ as the *Cross-Modal Scale* and $\theta'$ as the *Cross-Modal Parameter*. After training, we merge the model by computing and saving $\hat{\theta} = \theta + \lambda\theta'$ so that the model will no longer have extra parameters and the inference costs and model size will be identical to a regular model.

With Cross-Modal Re-parameterization, we equivalently realize the proposed M2PT transformer block with marginal training costs and completely no inference costs. For a linear layer whose parameters form a matrix $W \in \mathbb{R}^{D_{in} \times D_{out}}$ and the inputs and outputs are matrices $x \in \mathbb{R}^{B \times D_{in}}$ and $y \in \mathbb{R}^{B \times D_{out}}$. We omit the bias term for brevity and the original operation is formulated by

$$y = xW .  \qquad (6)$$

As described in Fig. 2, the linear layer and its counterpart take the same input. The output will be

$$y = xW + \lambda(xW') .  \qquad (7)$$

Note

$$xW + \lambda(xW') = x(W + \lambda W') ,  \qquad (8)$$

so that the two layers can be equivalently implemented by a single layer that has a trainable scalar $\lambda$ and an additional trainable matrix which is initialized with the counterpart in the auxiliary model. Both the original weight matrix and the additional one are trainable. At each forward computation, the layer computes the equivalent weight matrix and then uses it to project the input, which is

$$y = x(W + \lambda W') .  \qquad (9)$$

After training, we merge the parameters by computing $\hat{W} = W + \lambda W'$ and save it only. For inference, we simply construct a regular linear layer and load $\hat{W}$.

In summary, to construct and use an M2PT with Cross-Modal Re-parameterization, we

- Construct the tokenizer and head according to the target modality.
- Construct the transformer blocks with Cross-Modal Re-parameterization. For each linear layer, except for the original weight matrix, we add an extra trainable weight matrix and initialize it with the corresponding one from a transformer trained on the auxiliary modality and add a trainable scalar parameter initialized with 0.
- Train the re-parameterized cross-modal model just like we train a regular model.
- After training, convert the trained model and save the converted one for inference.

## 3 EXPERIMENTS

### 3.1 SETUP

**Datasets**. For image recognition, we evaluate the models' performance on three representative image datasets. 1) ImageNet-1K (Deng et al., 2009) is the most widely adopted benchmark for visual perception tasks, which contains nearly 1.3 million images of 1000 categories. 2) MSCOCO 2017 (Lin et al., 2014) is a common benchmark for object detection. M2PT is trained on the `train` set and evaluated on the `val` set with Mask RCNN (He et al., 2017). 3) ADE-20K (Zhou et al., 2017) is used for semantic segmentation experiments with UperNet (Xiao et al., 2018) and we adopt the single-scale evaluation setting. For point cloud, we evaluate the performance of M2PT on ShapeNet-Part (Yi et al., 2016), which contains 16,880 models and 16 categories. For audio recognition, following AudioMAE (Huang et al., 2022), we utilize the AudioSet-2k (Gemmeke et al., 2017) dataset. For video, we experiment on the action recognition dataset, Kinetics-400 (Kay et al., 2017), which contains 240k training videos and 20k validation videos categorized into 400 classes.

**Experimental details**. For a pair of target modality and auxiliary modality, we obtain the auxiliary model by self-supervised training on a dataset of the auxiliary modality. Specifically, the auxiliary image model is pretrained with MAE (He et al., 2022) on ImageNet-1K (Deng et al., 2009), the auxiliary point cloud model is pretrained with Point-MAE (Pang et al., 2022) on ShapeNet (Chang et al., 2015), the auxiliary audio model is pretrained with AudioMAE (Huang et al., 2022) on AudioSet-2M (Gemmeke et al., 2017), the auxiliary video model is pretrained with VideoMAE (Tong et al.,

Table 1: **Experimental results on image recognition tasks.** On ImageNet, we report the results with the linear layers in transformer blocks finetuned (tune acc) or fixed (fix acc). ∗: results are reported by running the official code. The architecture of every model is ViT-B.

| Method | ImageNet | | MS COCO | | ADE20K |
|---|---|---|---|---|---|
| | tune acc(%) | fix acc(%) | $AP_{box}$(%) | $AP_{mask}$(%) | mIOU(%) |
| **Pretrained setting** | | | | | |
| SemMAE(Li et al., 2022a) | 83.4 | 65.0 | - | - | 46.3 |
| MFF (Liu et al., 2023) | 83.6 | 67.0 | 48.1 | 43.1 | 47.9 |
| MAE∗(He et al., 2022) | 83.3 | 65.6 | 47.3 | 42.4 | 46.1 |
| M2PT-Video (Ours) | **83.6** ↑ 0.3 | **67.1** ↑ 1.5 | - | - | - |
| M2PT-Audio (Ours) | **83.7** ↑ 0.4 | **67.3** ↑ 1.7 | - | - | - |
| M2PT-Point (Ours) | **83.9** ↑ 0.6 | **67.8** ↑ 2.2 | **50.0** ↑ 2.7 | **44.0** ↑ 1.6 | 47.9 ↑ 1.8 |
| **From-scratch setting** | | | | | |
| ViT (Dosovitskiy et al., 2021) | 76.5 | 14.5 | 46.2 | 40.5 | 39.7 |
| M2PT-Point (Ours) | **81.9** ↑ 5.4 | **19.5** ↑ 5.0 | **48.9** ↑ 2.7 | **42.2** ↑ 1.7 | **42.5** ↑ 2.8 |

2022) on Kinetics-700 (Kay et al., 2017). For the fairness and reproducibility, we use their official code for pretraining. We do not use supervised pretraining because we would like to eliminate the effects of labels in the pretraining datasets. In terms of the initialization of the target model, we adopt two settings. 1) The target model (*i.e.*, the parameters denoted by $W$ in Eq. 9) is initialized with the aforementioned weights pretrained with the self-supervised methods on the target modality. We finetune the M2PT model with the default finetuning configurations described by the corresponding pretraining methods. The baseline model is also initialized with the pretrained weights and finetuned with identical configurations so that this setting is referred to as the *pretrained setting* for brevity. 2) The target model is randomly initialized as usual, and we use the widely adopted training configurations to train the M2PT model. The baseline model is trained from scratch with identical configurations for fair comparisons so that the setting is referred to as the *from-scratch setting* for brevity. In other words, the M2PT and the baseline models both have no weights pretrained on the target modality under this setting.

## 3.2 MAIN RESULTS

**Image recognition.** We first conduct a group of experiments under the pretrained setting, where the target weights are initialized with a ViT pretrained with MAE on ImageNet, and the auxiliary weights are from the models pretrained on video, audio, and point datasets, respectively. Such three models, which are labeled as M2PT-Video, M2PT-Audio, and M2PT-Point, respectively, and the baseline (the original MAE-pretrained ViT) are trained on ImageNet with the finetuning configurations originally adopted by MAE (He et al., 2022), and the resultant accuracies are reported in the "tune acc" column in Table 1. Then we transfer the best-performing model, which is M2PT-Point, to COCO object detection and ADE20K semantic segmentation tasks. The improvements are significant: on ImageNet, COCO, and ADE20K, the accuracy, box AP, and mIoU are improved by 0.6%, 2.7%, and 1.8%, respectively.

Apart from finetuning the target and auxiliary weights, we test another setting where the parameters of linear weights in transformer blocks are fixed, and only the Cross-Modal Scales together with the classifier are trainable. The accuracies are reported in the "fix acc" column. Naturally, under this setting, the baseline should be the MAE-pretrained ViT where only the classifier is trainable. Impressively, the improvement increases to 2.2% (67.8% vs. 65.6%), demonstrating that the weights obtained from the auxiliary modality work on another modality, even when the weights are fixed.

On the other hand, under the from-scratch setting, the baseline is a ViT trained from scratch, and the target weights of M2PT are also randomly initialized. The accuracy is drastically improved by 5.4% (81.9 vs. 76.5), suggesting the auxiliary weights significantly facilitate the training process. Intuitively, the Cross-Modal Scales are initialized with 0 but will soon become non-zero as the training proceeds so the model will be gradually influenced by the auxiliary weights and benefit

Table 2: **Experimental results on point cloud and audio recognition**. For point cloud analysis, we compare M2PT with PointNet++ (Qi et al., 2017), Point-BERT (Yu et al., 2022), and Point-MLP (Ma et al., 2022). For audio recognition, we compare with PSLA (Gong et al., 2021b), AST (Gong et al., 2021a), (Gong et al., 2022), and AudioMAE (Huang et al., 2022).

(a) Point cloud

| Method | ShapeNetPart | | PartNet |
|---|---|---|---|
| | mIoU$_C$ (%) | mIoU$_I$ (%) | mIoU (%) |
| **Pretrained setting** | | | |
| PointNet++ | 81.9 | 85.1 | 42.5 |
| Point-BERT | 84.1 | 85.6 | - |
| Point-MLP | 84.6 | 86.1 | 48.1 |
| Point-MAE | 84.2 | 86.1 | 47.4 |
| M2PT-Video | **85.6** ↑1.4 | **87.5** ↑1.4 | **50.1** ↑2.7 |
| M2PT-Image | **85.6** ↑1.4 | **87.5** ↑1.4 | **49.2** ↑1.8 |
| M2PT-Audio | **85.6** ↑1.4 | **87.5** ↑1.4 | **48.1** ↑0.7 |
| **From-scratch setting** | | | |
| N/A | 50.2 | 68.4 | - |
| M2PT-Video | **50.8** ↑0.6 | **68.8** ↑0.4 | - |

(b) Audio recognition

| Method | Model | Top-1 Acc. (%) |
|---|---|---|
| **Pretrained setting** | | |
| PSLA | CNN+Trans | 31.9 |
| AST | ViT-B | 34.7 |
| SSAST | ViT-B | 31.0 |
| AudioMAE | ViT-B | 35.3 |
| M2PT-Point | ViT-B | **35.6** ↑0.3 |
| M2PT-Video | ViT-B | **35.5** ↑0.2 |
| M2PT-Image | ViT-B | **35.6** ↑0.3 |
| **From-scratch setting** | | |
| N/A | ViT-B | 11.0 |
| M2PT-Point | ViT-B | **11.4** ↑0.4 |

from the modality-complementary knowledge. When we transfer such two models to COCO and ADE20K, we observe consistent improvements of +1.7% AP and +2.8% mIoU.

**3D Point Cloud Understanding.** Table 2a presents the experimental results on ShapeNetPart and PartNet datasets, where we compare M2PT with existing point cloud pretraining methods such as Point-BERT (Pang et al., 2022) and Point-MAE (Yu et al., 2022). M2PT brings out +1.4 class mIoU and +1.4 instance mIoU. Similarly, under the from-scratch setting, we observe +0.6% and +0.4% improvements in the class and instance mIoU, respectively.

**Audio Understanding.** For the pretrained setting, the target weights are initialized with an AudioMAE-pretrained model. As shown in Table 2b, we compare M2PT with existing methods including state-of-the-art pretraining-based methods SSAT and AudioMAE. M2PT brings out +0.3% top-1 accuracy on the Audioset balanced split, demonstrating that M2PT is also effective in audio recognition. Under the from-scratch setting, M2PT can bring out an improvement of +0.4%.

**Video Understanding.** For the experiments on Kinetics-400, we adopt only the pretrained setting because it is not a common practice to train a model from scratch on a video dataset, which would deliver inferior performance. We use the Video-MAE-pretrained ViT to initialize the target weights. Naturally, the baseline should be the VideoMAE-pretrained model directly finetuned on Kinetics-400. Table 3 shows that compared with previous works including SlowFast (Feichtenhofer et al., 2019), MViTv2 (Li et al., 2022b), TimeSFormer (Bertasius et al., 2021), and VideoMAE (Tong et al., 2022),

Table 3: **Experimental results on the Kinetics-400 datasets**.

| Method | Model | Top-1 Acc. (%) |
|---|---|---|
| SlowFast-101 | ResNet-101 | 79.8 |
| MViTv2-B | ViT-B | 81.2 |
| TimeSFormer | ViT-B | 80.7 |
| VideoMAE | ViT-B | 81.5 |
| M2PT-Point | ViT-B | **82.3** ↑0.8 |
| M2PT-Image | ViT-B | **82.2** ↑0.7 |
| M2PT-Audio | ViT-B | **82.2** ↑0.7 |

M2PT outperforms by at least +0.8% top-1 accuracy, which reveals that the temporal awareness for video understanding can also be enhanced with irrelevant data from other modalities.

## 3.3 ABLATION STUDIES

We evaluate the design choices of M2PT separately through a group of ablation studies under the pretrained setting on ImageNet and the auxiliary modality is point cloud.

**Applying Cross-Modal Re-parameterization to every linear layer delivers the best performance**. In each transformer block, we may choose to apply our method to any of the Query/Key/Value/projection layers in the attention block and the two linear layers in the FFN. Table 4 shows that changing any one of the layers brings improvements, and the best result is achieved by changing them all.

Table 4: **Ablation studies** on design choices of M2PT including the layers to re-parameterize and configurations of Cross-Modal Scale $\lambda$. The target dataset is ImageNet-1K and the auxiliary modality is point cloud.

| Components | | | | Cross-Modal Scale | | Top-1 accuracy (%) |
|---|---|---|---|---|---|---|
| `Attn QKV` | `Attn Proj` | `FFN 1st` | `FFN 2nd` | *Init.* | *Trainable* | |
| ✔ | | | | 0 | ✔ | 83.4 |
| | ✔ | | | 0 | ✔ | 83.6 |
| | | ✔ | | 0 | ✔ | 83.6 |
| | | | ✔ | 0 | ✔ | 83.7 |
| ✔ | ✔ | ✔ | ✔ | 0 | ✔ | **83.9** |
| ✔ | ✔ | ✔ | ✔ | $10^{-2}$ | ✘ | 83.5 |
| ✔ | ✔ | ✔ | ✔ | $10^{-2}$ | ✔ | 83.6 |
| ✔ | ✔ | ✔ | ✔ | $10^{-4}$ | ✔ | 83.6 |
| ✔ | ✔ | ✔ | ✔ | $10^{-6}$ | ✔ | 83.7 |

**Cross-Modal Scale should be initialized with 0**. By default, we initialize the Cross-Modal Scale $\lambda$ with 0 for every layer. We observe that initializing it to a higher value degrades the performance, suggesting that the initial state of the M2PT should be identical to the target weights (i.e., the weights pretrained with MAE, in this case).

**Cross-Modal Scale should be learnable**. Fixing the Cross-Modal Scale turns out to degrade the performance, suggesting that it is important to let the model learn how to combine the target weights and the corresponding auxiliary weights.

## 3.4 EMPIRICAL DISCUSSIONS

### 3.4.1 INVESTIGATING THE MODALITY-COMPLEMENTARY KNOWLEDGE

The observed improvements on multiple modalities have shown that the auxiliary transformer has learned some knowledge that is able to transfer to the target modality. We continue to investigate the properties of such modality-complementary knowledge through two groups of experiments (Table 5).

Table 5: ImageNet accuracy with changed order of auxiliary weights or fewer pretraining epochs.

| Order of aux weights | Epochs pretrained | Top-1 acc |
|---|---|---|
| Normal | 20 | 83.55 |
| Normal | 220 | 83.69 |
| Normal | 300 | 83.93 |
| Reversed | 300 | 83.61 |

1) We investigate if such knowledge is related to the ability to generally process hierarchical representations. Abstraction hierarchy exists in multiple modalities with concepts ranging from low-level to high-level, which may explain the transferability of the learned knowledge. For example, in image and point cloud, this hierarchy may include textures (in image) or individual points (in point cloud), object parts, and whole objects. Considering that the conceptual level a transformer block works on is determined by its depth, we design an experiment by reverting the order of the auxiliary weights. Specifically, the counterpart of the 1st target block should be the 1st auxiliary block, whose weights are connected via Cross-Modal Re-parameterization, which is obvious. However, since the transformer has 12 blocks, in this set of experiments we let the $i$-th block connect with the $(13 - i)$-th block so that the target-auxiliary correspondence is interrupted. We observe that doing so decreases the accuracy to 83.61%, which is 0.32% lower than the normal M2PT.

In summary, the modality-complementary knowledge in the auxiliary transformer can transfer to another modality but can be harmed if the low-to-high correspondence is interrupted, suggesting that such knowledge may help understand general hierarchical concepts regardless of the modality.

2) We investigate if a better pretraining process brings such knowledge of higher quality. We experiment using not well-trained weights as the auxiliary weights. Specifically, the default auxiliary weights are obtained through a 300-epoch self-supervised pretraining process on point cloud data, but we alternatively use the checkpoints saved at the 20th and 220th epoch, respectively, as the auxiliary weights. Not surprisingly, we observe that the performance degrades to 83.55% and 83.69%, respectively, which is still higher than the baseline.

This phenomenon suggests that the improvements brought by the auxiliary weights cannot be simply explained that the weights trained on another modality merely offer an initialization hardly better than the random initialization (if so, training the checkpoint at the 220th epoch to 300 epochs would not bring observable eventual improvements on the target modality).

### 3.4.2 DISCUSSION ON DATA SCALE

Previous works such as Image2Point (Xu et al., 2022) and Point-CLIP (Zhang et al., 2022) follow a common consensus that the modality owning a larger data scale could be utilized to benefit the other modality owning a smaller one. Therefore, Image2Point introduces image-pretrained models to data-insufficient 3D perception tasks. Differently, M2PT sets up a brand new methodology and breaks the former consensus: we discover that *even though the data scale of point clouds is limited, such data still brings out impressive improvements to the image, video, and audio perception tasks.* Impressively, The pretraining data of the latter modalities is larger in magnitude than that of point cloud, but point cloud makes a difference. The effectiveness of M2PT inspires us that for 3D vision research, which lacks huge-scale data for pretraining, M2PT introduces a promising direction to leverage irrelevant large-scale data from other modalities.

## 4 RELATED WORK

**Pretraining methods**. The evolution of unimodal pretraining paradigms has transitioned from supervised to self-supervised paradigms. For instance, (Devlin et al., 2019) introduced the mask-reconstruction paradigm and achieved remarkable outcomes. At that time, visual pretraining largely emphasized contrastive learning (Chen et al., 2020; He et al., 2020; Caron et al., 2021). Subsequently, leveraging the vast amounts of unlabeled data, the BERT paradigm gained traction and pioneers like MAE (He et al., 2022) successfully applied it to visual pretraining, while others (Pang et al., 2022; Gong et al., 2021a; Tong et al., 2022) extended this paradigm to areas like point cloud, audio, and video perception. On the other hand, multimodal pretraining methods Wang et al. (2021a;b; 2022a;b) are developing rapidly, which commonly uses the cross-attention mechanism as a cornerstone or large-scale paired cross-modal data (Radford et al., 2021).

In contrast, our method does not require any pairwise data but realizes cross-modal improvements with the original model structure which only comprises unimodal self-attention blocks.

**Structural Re-parameterization** is a methodology that constructs extra structures (e.g., convolutional layers) during training and converts the trained structures via transforming the parameters. A primary drawback of Structural Re-parameterization is that the constructed layers must participate in the forward and backward computations, resulting in significant extra training costs.

In contrast, Cross-Modal Re-parameterization is a simple re-parameterization method so that the only extra computation in the forward computation is adding up two weight matrices.

## 5 CONCLUSION AND LIMITATION

This paper aims to explore the feasibility and advantages of improving a model's performance on a specific modality with irrelevant data from other modalities. We propose a general framework named Multimodal Pathway and a concrete inference-cost-free implementation named Cross-Modal Re-parameterization. Multimodal Pathway represents an early exploration in this direction, which offers a novel perspective. We realize significant and consistent improvements on four representative modalities, which demonstrates the potential of our method as a promising approach.

The primary limitation is that the theory behind the improvements remains to be revealed. Apart from empirical explanations, we believe further investigations (e.g., a mathematically provable bound) will be useful, which may require a deeper understanding of the black box of deep neural networks.

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
