# OpenReview forum: "Multimodal Pathway: Improve Transformers with Irrelevant Data from Other Modalities"
_ICLR.cc/2024/Conference — ICLR 2024 Conference Withdrawn Submission_

### Official Review · Reviewer_BGNu · 2023-10-28

**Soundness:** 3 good
**Presentation:** 3 good
**Contribution:** 2 fair
**Rating:** 3
**Confidence:** 4

**Summary:**

This work tackles the problem of how to improve the transformer of one modality from the models of other modality with irrelevant data. It proposes a novel method named Multimodal Pathway equipped with cross-modal re-parameterization. It performs experiments with four modalities – images, videos, point clouds and audio with the datasets.

**Strengths:**

1. This work addresses an interesting and important problem – how to transfer knowledge from one modality to another modality.

2. The proposed approach is sound and reasonable.

**Weaknesses:**

1. The problem and motivation that this work focuses are not so novel.
- Many recent models do not require paired multimodal data for pre-training and fine-tuning.
- Knowledge transfer from one modality to another is actively studied in many directions.
- Unfortunately, the related work of this paper is largely cursory.
- Many previous works have shown that the four modalities that this work considers – images, videos, point clouds and audio – are related enough to learn from one modality and help for another modality with no pairing.

2. The effectiveness of the proposed approach is not sufficiently demonstrated in experiments.
- The reported performance improvements are somewhat marginal over MAE as shown in Table 1 (i.e., mostly less than 1.0% in accuracy and at best <2.0%).
- Overall, the proposed cross-modal re-parameterization seems reasonable, more thorough experimental supports may be required.
- More supports include more baselines, other novel modalities, more performance gaps, etc, which can make this submission much stronger.

**Questions:**

Please refer to the weakness.

---

> ### Author Response · Authors · 2023-11-11
> **Response to Reviewer BGNu (weakness 1)**
>
> First of all, we would like to respectfully thank the reviewer's time and effort in evaluating our work. However, we would like to clarify that some of the mentioned weaknesses seem to be derived from misunderstandings. We address these concerns below and hope that our explanations will lead to a reconsideration of the paper's evaluation.
>
> **"Many recent models do not require paired multimodal data for pre-training and fine-tuning."**
>
> The advantages of multimodal pretraining have been widely adopted by the community, such as vision-language pretraining [CLIP, BEIT, Flamingo], and vision-language-audio pretraining [VALOR, VAST, UnIVAL]. Some methods do not use paired data but interleaved data like Flamingo (e.g., a text paragraph together with several images describing objects mentioned in the text), so that the data is still relevant. In this paper, we mentioned prior works with **paired or interleaved data** (e.g., in Figure 1). We will add more detailed discussions in the revised version. However, besides these successful attempts with multimodal relevant data, multimodal pretraining with completely irrelevant data, which is **neither paired nor interleaved**, especially with image, point cloud, audio, and video data, remains under-explored. Therefore, the existing multimodal pretraining works do not undermine the novelty of our work, and we are confident that our work will be inspiring and meaningful for the research area.
>
> **"Knowledge transfer from one modality to another is actively studied in many directions."**
>
> We understand that the reviewer is familiar with multimodal knowledge transfer. However, these works are mainly focused on transferring knowledge from data-sufficient modalities such as images to data-insufficient modalities such as point cloud or audio. However, bidirectional multimodal knowledge transfer is not mentioned before. Meanwhile, it's the first time to use data-insufficient modality such as point cloud to assist data-sufficient modality such as image in our paper. The proposed method Multimodal Pathway even achieves impressive improvements (e.g. +0.6% on ImageNet, +2.2% on ImageNet Linear Probing, +2.7% on MS COCO object detection, and +2.8% on ADE20K segmentation. )
>
> **Unfortunately, the related work of this paper is largely cursory.**
>
> We thank the reviewer for the constructive suggestions. We discussed pretraining methods and re-parameterization techniques because they are closely related to our method. If the reviewer could name a few significant references lost in this paper, we are willing to add a thorough discussion.
>
> **"Many previous works have shown that the four modalities that this work considers – images, videos, point clouds and audio – are related enough to learn from one modality and help for another modality with no pairing."**
>
> To the best of our knowledge, we are the first to conduct research on this topic. We sincerely hope that the reviewer can help us with more relevant references.
>
> (please see our response to Weakness 2 in the next comment)

---

> ### Author Response · Authors · 2023-11-11
> **Response to Reviewer BGNu (weakness 2)**
>
> (This is the second response. Please see our response to Weakness 1 in the last comment)
>
> **"The reported performance improvements are somewhat marginal over MAE as shown in Table 1 (i.e., mostly less than 1.0% in accuracy and at best <2.0%)."**
>
> As we introduced previously, Multimodal Pathway achieves impressive improvements (e.g. +0.6% on ImageNet, +2.2% on ImageNet Linear Probing, +2.7% on MS COCO object detection, and +2.8% on ADE20K segmentation.) Please refer to the experimental results in Table 1.
>
> Moreover, we would like to highlight that MAE is a powerful pretraining method, and it is challenging to gain further improvements on top of it. Some recent works aimed to improve MAE, e.g., SemMAE(Li et al., 2022a, published at NeurIPS) achieved ImageNet accuracy of 83.4 (+0.1), MFF (Liu et al., 2023, published at ICCV) achieved 83.6 (+0.3), which are cited in Table 1. Our method achieved an accuracy of +0.6, which can be seen as significant.
>
> **"Overall, the proposed cross-modal re-parameterization seems reasonable, more thorough experimental supports may be required. More supports include more baselines, other novel modalities, more performance gaps, etc, which can make this submission much stronger."**
>
> We thank the reviewer for the constructive suggestion. In our manuscript, we have conducted experiments for **4** modalities and constructed **12** types of multimodal pathways on **7** widely adopted benchmarks. More experimental results can be easily found in our paper Table 2 and Table 3. Meanwhile, to fairly compare our method with existing advanced pretraining methods, we take MAE, Point-MAE, AudioMAE, and Video MAE as baselines in these experiments, our methods bring consistent improvements, and 12 combinations between 4 modalities can effectively be improved with multimodal pathway.
>
> **In summary**, we appreciate the reviewer's comments and hope that our clarifications on the misunderstandings will address the concerns.
>
> We kindly request the reviewer to reconsider the evaluation in light of these explanations.
>
> Please let us know if there is anything we can do to convince the reviewer to raise the score.

---

### Official Review · Reviewer_8Bi6 · 2023-10-31

**Soundness:** 2 fair
**Presentation:** 2 fair
**Contribution:** 2 fair
**Rating:** 3
**Confidence:** 4

**Summary:**

This work proposes Multimodal Pathway Transformer (M2PT), a model to improve target modality from other modalities with non-paired data. M2PT consists of modality-specific tokenizers to transform raw inputs into features, and multiple linear layers inside each transformer block. Given the non-paired data of different modalities,  M2PT processes these data simultaneously and shows that the auxiliary data can improve the model's performance on target modalities.

**Strengths:**

1. This method is simple and the paper is easy to understand.
2. Lots of experiments are conducted to show that auxiliary modality can improve the model's performance on target modality.

**Weaknesses:**

1. The author claims that incorporating the auxiliary modality would improve the model's performance on the target modality, even if there is no any relevance between the data. However, the reasoning behind this enhancement in performance remains unexplained. Furthermore, it cannot be confirmed that non-paired data is entirely unrelated, thus raising doubts about the validity of all related statements.

2. In M2PT, numerous additional parameters were introduced, which could potentially account for the observed model improvement, rather than the inclusion of an auxiliary modality. The author didn't conduct any ablation experiments to investigate this.

**Questions:**

see weakness above

---

> ### Author Response · Authors · 2023-11-11
> **Response to Reviewer 8Bi6 (weakness 1)**
>
> We thank the reviewer for their efforts and valuable feedback. However, we would like to clarify that some of the mentioned weaknesses seem to be derived from misunderstandings. We address these concerns below and hope that our explanations will lead to a reconsideration of the paper's evaluation.
>
> Weakness 1. **"It cannot be confirmed that non-paired data is entirely unrelated"**
>
> **Clarification**: We respectfully disagree with this comment. We are confident that the datasets we use are unrelated. For example, the reviewer may be concerned that the image dataset contains images of cars, while the audio dataset contains sounds from cars. Although people may perceive these as related due to their learned concepts, the model has no such concepts. From the model's perspective, the audio and images are completely unrelated. Furthermore, to ensure irrelevance, we avoid using labels during pretraining and employ an MAE-style pretraining. Consequently, without any labels, the model does not even know that the sounds are "sounds from cars" or that the images are "cars," so it cannot establish any relation between the images and audio.
>
> **the reasoning behind this enhancement in performance remains unexplained**
>
> Though completely explaining the phenomena (i.e., transformer benefits from irrelevant data from another modality) is beyond the scope of this paper, which may require the research community to gain a deeper understanding of the black box of deep learning. We have some preliminary explanations and intuitions - some modality-agnostic knowledge, which is about general sequence-to-sequence modeling, exists in transformers. We stated this in the paper ("In other words, apart from the obvious modality-specific knowledge acquired through training on a specific modality, we seek the modality-complementary knowledge of sequence-to-sequence modeling in transformers and will show that it does exist"). We understand that the reviewer may concern that the specific form and emergence of such  "modality-agnostic knowledge" also needs to be explained, which are discussed as follows.
>
> For example, while a transformer is being pretrained with MAE, it learns both (ability A) how to understand and reconstruct the image and (ability B) how to generally transform the tokens from the lower-level patterns to a higher level without assuming they originally come from images. Meanwhile, as another transformer is being pretrained with audio data, it learns both the different "ability A" for audio and similar "ability B".
>
> Our experiments verified that "ability B" does exist. But it is difficult to formally define it. We reckon that it should be the ability to capture and understand the common structures of data of all modalities - the abstract hierarchy. A model for image recognition gradually extracts textures from lines, colors, and angles, then recognizes components of objects from textures and shapes. Similar hierarchical structures exist in data of other modalities, and the ability to extract information of higher hierarchy from lower-level tokens on a certain modality should be similar to the ability on another modality.
>
> Generally, this paper is aimed at showing these intriguing phenomena and a simple yet effective method as pioneering research. Though formally defining and mathematically proving the concepts seem infeasible for the research community's latest understanding of the black box nature of deep learning, we believe it is of vital importance to make the community aware of such phenomena and our early exploration, which may deepen the community's understanding of transformers, pretraining, and multimodal learning.
>
> (please see our response to Weakness 2 in the next comment)

---

> ### Author Response · Authors · 2023-11-11
> **Response to Reviewer 8Bi6 (weakness 2)**
>
> (This is the second response. Please see our response to Weakness 1 in the last comment)
>
> Weakness 2.**"Numerous additional parameters were introduced"**
>
> **Clarification**: We would like to clarify two points: **(1)** We do not introduce any extra structures or parameters into the resultant model. We only temporarily use some additional parameters during training, which are merely aimed at changing the parameterization of the existing parameters. Such parameters can be equivalently eliminated so that the resultant model has the same number of parameters and identical structure to the original model, with the only difference being its improved performance. **(2)** Even though we use additional parameters only during training, we have verified that the performance improvements are not merely due to the more training-time parameters. The reviewer commented that we did not conduct any experiments to investigate this, which is not true.
>
> **(1) We do not introduce any extra structures or parameters into the resultant model.**
>
> The additional parameters during training are merely a temporarily changed parameterization of the existing parameters. To make this clear, we first explain the concept of parameterization in neural networks. For example, given an arbitrary fully connected layer (FC) in a neural network, assume its numbers of input features and output features are C and D, respectively, it requires C x D parameters to finish its computation (linearly mapping the C-dimensional input to D-dimensional output). We refer to such C x D parameters as its essential parameters.
>
> **Vanilla parameterization**: The most natural way to represent C x D parameters is using a C x D matrix, so "using a matrix to represent the parameters of an FC layer" is referred to as a "parameterization". Since it is the most naive manner, we refer to it as the vanilla parameterization. In this parameterization, let x and y be the inputs and outputs and W be the matrix, respectively, the computation of this FC layer can be represented by y=xW.
>
> **Our proposed parameterization**: The computation of this FC layer is changed to y=x(W+λW′). In other words, the parameterization is changed from "using a matrix to represent C x D parameters" to "using two matrices and a scalar to represent C x D parameters". This can be seen as a method to **"merge" the auxiliary model into the current model being trained**. But we would like to note that the FC layer still has the same number of essential parameters (since it still maps the same inputs to the same shapes of outputs), and only the parameterization changes. This parameterization is designed in this way because we want to equivalently implement the block shown in the middle of Figure 2. The reviewer may concern that the training-time model will require 2x layers and computations as the original model, which seems to be indicated by the middle figure in Figure 2. While, as we explained in the Introduction and caption of Figure 2, it is merely an abstract high-level idea, which is **efficiently** and **equivalently** realized by merely changing the parameterization (the right figure in Figure 2). In summary, the model being trained has the same number of linear layers as the original model, and **each layer merely needs to compute W+λW′ before linearly mapping x to y**.
>
> **Convert parameterization**: After training, the eventual weight matrix W^ is derived by W+λW′ so that the parameterization becomes the same as the original model. We only save this matrix. For inference, we simply construct a regular model and load the converted weights, so our method does not introduce additional parameters.
>
> **(2) Even though we use additional parameters only during training, we have verified that the performance improvements are not merely due to such extra parameters.**
>
> The results are reported in Table 5 (ImageNet accuracy with changed order of auxiliary weights or fewer pretraining epochs). We observe that changing the order of auxiliary weights and using not fully trained auxiliary weights result in lower accuracy than the fully trained auxiliary weights. If the performance improvements were merely due to more training-time parameters, using more parameters—even if they were not fully trained or randomly initialized—would result in the same performance. Since this is not the case, we can confirm that the performance improvements are not merely due to more training-time parameters.
>
> We will enhance the description of relevant sections in the revised version. For example, in Section 3.4.1, we will further analyze the results and add the discussions above.
>
> We appreciate the reviewer's constructive feedback and hope that our clarifications on the misunderstandings will address the concerns.
>
> We kindly request the reviewer to reconsider the evaluation in light of these explanations.
>
> Best Regards,
> Authors

---

### Official Review · Reviewer_hWyf · 2023-10-31

**Soundness:** 3 good
**Presentation:** 3 good
**Contribution:** 3 good
**Rating:** 6
**Confidence:** 2

**Summary:**

This paper proposes a cross-modal re-parameterization method to investigate the usage of irrelevant data to improve the overall performance of models.

**Strengths:**

This work present a wide study on multiple modalities of data, as well as tasks, which I think it's a contribution to the community. Moreover, the cross-modal reparameterization seems simple and straightforward yet effective, compared to largely pretrained MAE. Overall, the paper is well-written and easy to follow.

**Weaknesses:**

It would be great or to visualize the intermediate representation/weights with or without the re-parameterization method to see how it shifts. Also, does the re-parameterization also help the performance of the irrelevant dataset?

**Questions:**

1. Can the authors give a rationale or even guess why including irrelevant data works?

---

> ### Author Response · Authors · 2023-11-11
> **Response to Reviewer hWyf**
>
> Thank you for appreciating the novelty, simplicity, and effectiveness of our method!
>
> We did not visualize the intermediate representation or weights because there is yet no widely accepted manner to visualize linear layer's weights or tokens in transformers. In fact, such visualization tools usually make sense with some input samples but show almost no changes with some other samples, and we do not want to do cherry-picking. The answer to the first question "does the re-parameterization also help the performance of the irrelevant dataset" is yes, because the improvements are bi-directional, as shown in Figure 3. For example, image improves point cloud b 1.8, and ponit cloud improves image by 0.6.
>
> Answer to the question:
>
> Yes, as we stated in the paper ("In other words, apart from the obvious modality-specific knowledge acquired through training on a specific modality, we seek the modality-complementary knowledge of sequence-to-sequence modeling in transformers and will show that it does exist"), we verified that some modality-agnostic knowledge, which is about general sequence-to-sequence modeling, exists in transformers.
>
> For example, while a transformer is being pretrained with MAE, it learns both (ability A) how to understand and reconstruct the image and (ability B) how to generally transform the tokens from the lower-level patterns to a higher level without assuming they originally come from images. Meanwhile, as another transformer is being pretrained with audio data, it learns both the different "ability A" for audio and similar "ability B".
>
> Our experiments verified that "ability B" does exist. But it is difficult to formally define it. We reckon that it should be the ability to capture and understand the common structures of data of all modalities - the abstract hierarchy. A model for image recognition gradually extracts textures from lines, colors, and angles, then recognizes components of objects from textures and shapes. Similar hierarchical structures exist in data of other modalities, and the ability to extract information of higher hierarchy from lower-level tokens on a certain modality should be similar to the ability on another modality.
>
> However, though we have demonstrated that such an ability does exist, we cannot mathematically define it due to the black-box nature of deep learning. In fact, almost every concept we used in the paragraph above has no formal definition in the deep learning literature (e.g., we are still unable to calculate the degree of hierarchy given the values of tokens or ensure if a feature is on a higher level than another feature). Generally, this paper is aimed at showing these intriguing phenomena (i.e., transformer benefits from irrelevant data from another modality) and a simple yet effective method as pioneering research. Completely explaining the phenomena is beyond the scope of this paper, which may require the research community to gain a deeper understanding of the black box of deep learning.
>
> Thank you again for appreciating our work, and we would be grateful if you could raise the score.
>
> Please let us know if there is anything we can do to convince you further raise the score.